# Research on the Effect of Regional Talent Allocation on High-Quality Economic Development—Based on the Perspective of Innovation-Driven Growth

**Lu Liu [1], Shenshen Si [2,*] and Jing Li [1]**

[1] School of Economics, Anhui University, Hefei 230601, China
[2] College of Finance, Anhui University of Finance and Economics, Bengbu 233030, China
* Correspondence: shenshen@aufe.edu.cn

**Abstract:** As China's economy moves towards a stage of high-quality development and shifts its economic development goals from GDP growth to green and sustainable growth, technological support is essential for achieving green and sustainable economic growth. Therefore, the supply of talent, as the source of innovation, is crucial. Against the backdrop of relying on innovation to drive high quality economic development, achieving the effective allocation of talent within a spatial range to maximize the release of human capital dividends and promoting the benign interaction between talent regional allocation and technological innovation is an urgent issue that needs to be addressed to achieve environmentally sustainable economic development. Based on this, this paper studies the effect of regional talent allocation on high-quality economic development, reveals the impact mechanism of regional talent allocation on high-quality economic development, and uses the panel data of 258 cities in China from 2004 to 2019 to empirically test the impact of regional talent allocation on high-quality economic development, with a view to improving regional talent allocation, releasing talent potential, and promoting the improvement of regional environmental quality and the convergence of new ideas for high-quality economic development. This research indicates the following: (1) The improvement of the talent regional allocation level can effectively promote high-quality economic development, and mechanism verification shows that talent regional allocation promotes high-quality economic development by influencing regional innovation;. (2) The heterogeneity test found that the impact of regional talent allocation on high-quality economic development indicated a law of an increasing marginal effect from east to west, while innovation drive and the interaction between regional talent allocation and innovation drive showed the strongest characteristics in the central region, followed by the west, with the weakest being in the east. In addition, both the regional allocation of talent and the innovation-driven impact on the high-quality development of the economy have a higher marginal effect in non-urban agglomeration cities than in urban agglomeration cities. (3) There is a dual threshold effect of innovation-driven regional talent allocation on the development of a high-quality economy. When the innovation drive is between 0.4898 and 10.2214, the spillover effect of innovation-driven talent flow is less than the negative impact of talent flow, which is not conducive to the development of a high-quality economic development effect of regional talent allocation. Studying the impact of regional talent allocation on high-quality economic development not only helps to supplement and improve the theory of human capital mobility, providing new explanations for high-quality economic development in the new era, but also contributes to enriching the content of modern macroeconomic theory.

**Keywords:** high-quality economic development; regional allocation of talent; innovation driven; green development; sustainable

## 1. Introduction and Literature Review

The rapid growth of China's economy in the past three decades has raised discussions revolving around environmental protection requirements. With the transformation of

China's economy from a stage of rapid growth to a stage of high-quality development, Chinese local governments have prioritized the development model to a green, sustainable format [1]. The green and sustainable development of the economy cannot be separated from the support of technology [2]. As the core element of scientific and technological innovation, talent has increasingly become an important tool for local governments to develop the economy [3], and the effective allocation of talent can guide and coordinate the utilization of the resources of the whole country [4]. In the era of a planned economy, the free flow of talent resources was restricted, and talented individuals could not choose suitable positions according to their own preferences, which caused serious talent misplacement and restricted the productivity of resource elements, resulting in the increasingly prominent contradiction between economic development and environmental degradation [5]. Since the reform and opening up of China, and with the gradual opening up of the labor market, the mobility of the labor force in different departments has been increasing, and the allocation of talent has also improved. The improvement of talent allocation efficiency among departments has provided important support for the rapid growth of China's economy for more than 30 consecutive years [6]. It is undeniable that, although the talent allocation has been greatly improved and optimized with the gradual opening up of the labor market [7], the serious problem of talent misplacement still exists, and shows a certain degree of the "off the real to the virtual" tendency [8–10]. The effective allocation of talent is related to the improvement of environmental quality and the smooth progress of high-quality economic development [11]. Under the background of relying on innovation to drive high-quality economic development [12], achieving high-quality economic development with the goal of environmental protection is an urgent problem to be solved in order to realize the effective spatial allocation of talent, to maximize the release of human capital dividends, and to realize the benign interaction between the regional allocation of talent and technological innovation [13,14]. Based on this, this paper studies the effect of regional talent allocation on high-quality economic development, with a view to gathering new ideas for improving regional talent allocation, releasing talent potential, and promoting regional environmental quality improvement and high-quality economic development in the new era.

Academics have carried out many studies on the topic of talent. In terms of the distribution of talent, Kniveton (2004) found that a large number of outstanding talented individuals in the college student employment market have gradually formed a preference for employment within the system, which, to some extent, has caused an imbalance in the allocation of talent within and outside the system [15]. At the industry level, China's primary industry talent supply is excessive and deteriorating, and the talent supply of life services and public services in the tertiary industry is relatively excessive, while the talent supply of secondary industries is becoming increasingly serious [16]. Ren and Sun (2021) estimated the degree of talent mismatch between industries and found that the degree of talent distortion mismatch between different industries in China is large [17]. In addition to the industrial misplacement of talent, there are also irrational phenomena in the regional distribution of talent in China. The agglomeration and siphon effects of large cities have caused a backlog of talent, and the excessive concentration of talent has led to the continuous decline in marginal output. Correspondingly, the "talent shortage" staged in medium and small cities, coupled with the strategy of strengthening the provincial capital in the central and western regions, has caused an excessive concentration of talent in the provincial capital cities, while small cities and ethnic minority areas in the central and western regions are gradually becoming talent depressions [18]. Xie (2019) conducted a study on the inter-provincial distribution of talent in China and found that the mismatch of talent in China is spatially heterogeneous, and the degree of human capital mismatch in the central and western regions is far less than that in the coastal eastern regions [19]. The research of Yang and Xie (2021) found that the difference in talent scale between different provinces in China is obvious due to the difference in initial endowments such as the spatial regions and economy in different provinces [20]. Li and Chen (2019) studied the regional



distribution of talent in China and found that there is a "Matthew effect" in the distribution of human capital in various regions [21]. In economically developed provinces, the scale of talent is relatively high, while in backward provinces, talent is scarce and difficult to supplement. The gap between developed and underdeveloped provinces is growing, and the polarization is becoming increasingly serious.

In terms of the role of talent, the new economic growth theory emphasizes the promotion of human capital accumulation in innovation [22,23]. The greater the talent accumulation in an economy, the higher the technological level of the economy, and the more able it is to promote the improvement of the environmental quality and the green and sustainable development of the economy [24]. However, this is not the case in reality. Rodríguez-Clare (2001) found that the technological level of some developed countries is higher than that of developing countries, but the level of the talent scale is lower than that of developing countries in some periods [25]. This transnational difference between talent stock and innovation level cannot be explained by the innovation theory of development economics, which shows that the contribution of talent to innovation is not only related to the accumulation level, but also the allocation level [26]. Some scholars have found that China's talent has reached a sufficient scale in quantity and improved greatly in quality, but its independent innovation capability has seriously lagged behind the actual needs of economic development, and its contribution to TFP cannot be reflected. The reason is that the effective allocation of talent in China deviates from the equilibrium [27,28]. An imbalance of talent allocation will lead to many adverse consequences. Lai and Ji (2015) pointed out that a large number of talented people are blocked in the monopoly departments, resulting in a lack of innovative talent in the market-oriented competition department, thus inhibiting the level of independent innovation of Chinese enterprises [29]. Li and Ma (2021) found that talent mismatch between monopoly and competition departments makes it difficult to carry out innovation activities and to upgrade the industrial structure [30]. On the other hand, it causes the polarization of income between the non-productive administrative monopoly department and the productive competition department. The deviation of talent allocation to the virtual economic sector will easily lead to an imbalance of the economic structure and increase the probability of the major risk of "shifting from reality to reality" [31,32].

With the gradual disappearance of labor cost advantages and investment growth momentum, and the diminishing effect of "learning by doing", the late-developing advantage of relying on factors to drive economic growth is no longer available, and the extensive growth model is increasingly difficult to sustain. The imbalance of talent distribution has seriously inhibited the release of talent potential, resulting in an insufficient regional allocation of talent, and it has become an obstacle to innovation-driven growth [33]. To realize the transition from factor- and investment-driven to innovation-driven growth, the rational allocation of talent is key. In terms of the relationship between talent and high-quality economic development, Sabadie (2014) pointed out that human capital is the key to improving regional competitive advantages, and also the core of improving environmental quality and achieving sustainable economic development [34]. Hao's (2021) research found that economic growth causes losses in environmental quality, but the improvement of the human capital level can improve the environment and promote green economic development [35]. Sarkodie (2020) pointed out that human beings are the catalyst for exacerbating environmental degradation [36], but Zhang's (2021) research found that high-quality human capital can improve environmental quality and promote high-quality economic development by promoting technological progress to reduce pollutant emissions [37].

After sorting through the existing literature, it was found that there are talent mismatches between departments [15], industries [16,17], and regions [18–20]. The research on the consequences of talent mismatch mainly focuses on talent mismatch innovation [27,28], industrial structure [31,32], income distribution [30], total factor production efficiency, and other aspects [27,28]. The role of human capital in environmental pollution [24,36] and green and sustainable economic development [34,35,37] has also attracted the attention of many scholars. Although some scholars have also paid attention to the impact of talent

allocation on economic growth [38], there is little research on the effect of talent allocation on high-quality economic development. At the critical juncture of China's economic development from a high-speed growth stage to a high-quality development stage, studying the impact of talent allocation on high-quality economic development is also of great practical significance for smoothly promoting the transformation of the economic growth mode and promoting high-quality economic development.

## 2. Theoretical Analysis

### 2.1. Functional Mechanism of Regional Talent Allocation on High-Quality Economic Development

Although China's technological level has been greatly improved in the process of economic catch-up for many years, there is still a gap between China and developed countries. China still needs to continue to explore independent innovation, and talent support is indispensable in the promotion of innovation activities, especially the support of innovative talent [39,40]. Inadequate regional allocation of talent will affect the development of innovation activities, which is not conducive to the improvement of innovation efficiency [27,28,41]. In the stage of innovation-driven development, various regions have increased their efforts to attract talent, which has promoted the flow of talent among different regions [42]. As an important channel to improve the regional allocation of talent, talent flow not only helps to encourage talented people to independently improve their skills to meet the requirements of the job, as well as solving the problem that some talent do not use what they learn, but also improves the matching efficiency of talent and technical jobs in the market. It is convenient for individuals of talent to accumulate innovation experience in the process of "learning by doing" [43], so as to accelerate the transformation of ordinary talent into innovative talent, accumulate talent reserves for the development of innovation activities, and promote regional independent innovation ability and the ability to absorb foreign technology, while improving the regional talent allocation level [44], so as to promote the improvement of regional environmental quality and high-quality economic development. The transformation of talent flow direction in the innovation-driven development stage is conducive to breaking the situation whereby there is more investment and less output in education in remote and backward areas, and improving the enthusiasm of local schools to increase the talent supply, so as to alleviate the problem of unbalanced regional talent allocation [45]. As high-end knowledge owners, the improvement of regional talent allocation is not only conducive to the exchange of heterogeneous knowledge, but is also able to promote knowledge innovation and creation, and the exchange and sharing of technical experience [41]. It can also effectively alleviate the efficiency gap caused by the distorted regional talent allocation, thus injecting new momentum into the improvement of environmental quality and high-quality economic development.

Since those in the labor force can work in almost any position, they will give priority to the jobs that enable them to obtain better welfare and development opportunities [46], meaning that the talent in the backward regions will be attracted by material incentives and job opportunities and flow to those regions with better economic development, resulting in insufficient talent allocation in the backward regions [47]. The shift in economic growth mode to innovation-driven growth in the stage of high-quality development is conducive to breaking this situation. In order to attract and retain talent, the government and enterprises need to guide the inflow of talent by raising salaries and other subsidies to provide sufficient economic opportunities, a pleasant cultural environment, and a rich learning atmosphere for talent [48]. This is conducive to the improvement of talent efficiency and the allocation level. The improvement of regional talent allocation has strengthened the connection between enterprises and promoted the formation of horizontal connection closely related to innovation activities and the gathering innovation model [49]. This has not only promoted the transformation and upgrading of traditional industries, but has also accelerated the rapid development of emerging industries such as information technology, the Internet, and new energy, so as to attract related enterprises and similar industries to converge in space, which is conducive to the role of the talent-gathering effect. In addition, innovation-driven

development can help guide the free flow of various production factors among different sectors, promote the integration and reallocation of capital, labor, and other production factors, and reduce the path dependence on the traditional economic development mode by increasing the input of innovation factors [50]. Therefore, in the process of China's economic growth mode changing from factor-driven to innovation-driven development, we should give full play to the leading role of talent allocation in innovation so as to promote the realization of the goal of high-quality economic development with environmental protection as the core [51,52]. Based on this, the following hypothesis is proposed:

**Hypothesis H1:** *Regional talent allocation can promote regional innovation and thus improve environmental quality and high-quality economic development.*

*2.2. Nonlinear Effect of Regional Talent Allocation on High-Quality Economic Development*

The core role of regional talent allocation in improving the environmental quality lies in the absorption, utilization, and transformation of new knowledge to promote technological progress [53]. If talents work in the same position for a long time, it will lead to their fixed thinking and lack of sensitivity to observing problems, and their creativity will decrease with the increase in the number of working years or even remain at a relatively low level compared to their peak performance [44]. As a result, talented people rely on consumption behaviors such as food, clothing, housing, and transportation and participate in low-end labor to stimulate economic growth. This allocation of talent is inefficient. As far as innovation drive is concerned, innovation drive stimulates the mobility of talent. Mobility can not only encourage talent to find the most suitable positions and improve the efficiency of regional talent allocation, but also broaden the channels of knowledge spillover [54] and accelerate the horizontal diffusion of knowledge and technology among industries and regions, thus improving the overall innovation ability and regional science and technology level, and accelerating the process of economic growth from factor-driven to innovation-driven growth, so as to reduce pollution emissions and boost high-quality economic development. In fact, enterprises will set up a certain inspection period for new talent after they flow to new jobs. During the inspection period, these talented individuals will not be given challenging jobs, and the display of their ability will be limited. When it is difficult for the spillover effect of talent flow to make up for the negative impact caused by mobility, it is not conducive to the improvement of the level of regional talent allocation. When the externality of talent flow is higher than the negative effect of mobility, it is conducive to the role of regional talent allocation in high-quality economic development [55]. Under the innovation-driven growth model, various policies have been introduced all over the country, pushing the cross-regional flow of talent to a climax. However, a moderate talent flow can intensify the competition in the labor market. The role of the survival of the fittest mechanism can not only motivate employees to improve their skills spontaneously, but also mobilize their learning motivation and even stimulate their innovation potential under the new environment, effectively promoting regional independent innovation capacity and foreign technology absorption capacity [44]. Therefore, the intensification of competition between regions caused by innovation drive will lead to talent flow. When the spillover effect of talent flow is greater than the negative impact, it is conducive to promoting the role of regional talent allocation in high-quality economic development. When innovation drive causes talent who do not need to flow to participate in the flow under external impact, it will place talent in the adaptation period of the new environment, resulting in a low efficiency of regional talent allocation, and inhibiting the role of regional talent allocation in high-quality economic development [47]; therefore, under different innovation drive levels, talent flow affects the effect of regional talent allocation on high-quality economic development [38,56]. Based on this, the following hypothesis is proposed:

**Hypothesis H2:** *The effect of regional talent allocation on high-quality economic development is nonlinear under the influence of innovation drive intensity.*

### 3. Model Construction and Indicator Selection

*3.1. Model Construction*

It can be seen from the above mechanism analysis that regional talent allocation can effectively promote regional innovation and promote high-quality economic development. Therefore, in order to identify the impact of regional talent allocation on high-quality economic development, this paper constructed the following econometric model:

$$ecogrq_{i,t} = \alpha_0 + \alpha_1 talalloc_{i,t} + \alpha_2 controls + \lambda_i + \phi_t + \varepsilon_{i,t} \qquad (1)$$

where $ecogrq_{i,t}$ refers to the level of high-quality economic development, $talalloc_{i,t}$ refers to the level of regional talent allocation, *controls* refers to the control variables, including industrial agglomeration level (*indclu*), marketization level (*market*), salary incentive level (*incent*), financial autonomy level (*fiauto*), and financial development level (*finance*), $\lambda_i$ refers to the individual effect, $\phi_t$ refers to the time effect, $\varepsilon_{i,t}$ refers to the random disturbance term, $t$ refers to time, and $i$ refers to region.

*3.2. Index Selection and Description*

3.2.1. Interpreted Variable

High-quality economic development (*ecogrq*): High-quality economic development is sustainable development that takes into account the comprehensive consideration of all levels of the economy and society. This paper referred to the measurement method of Guo et al. (2022) to build a high-quality economic development indicator system from the basic aspects of growth and social achievements [57]. In this system, the fundamentals of growth are reflected in four aspects: economic growth intensity, economic growth stability, economic growth rationalization (the rational use of economic growth (1-Theil index), where the formula for calculating the Theil index is ($TL = \sum\limits_{i=1}^{n} (\frac{Y_i}{Y}) \ln(\frac{Y_i}{L_i} / \frac{Y}{L})$), and economic growth extroversion; the social achievements are measured from human capital and ecological capital. First, every indicator is standardized, and then the principal component analysis method is used to determine the weight of the indicators at all levels, finally synthesizing the high-quality economic development indicators. The measurement methods of the indicators at all levels are shown in Table 1 below.

**Table 1.** Descriptive statistics of variables.

| Level I Indicators | Secondary Indicators | Measurement Method |
|---|---|---|
| Fundamentals of growth | Economic growth intensity | Regional real GDP per capita |
| | Stability of economic growth | Reciprocal coefficient of variation of GDP growth rate |
| | Rationalization of economic growth | 1-Theil index |
| | Extraversion of economic growth | Proportion of net exports in GDP |
| Social achievements | Human capital | Number of college students per 10,000 people |
| | Ecological capital | Ratio of GDP to PM2.5 concentration |

3.2.2. Core Variables

Regional allocation of talent (*talalloc*): The existing literature mainly uses the average number of years of education or the number of people with college degrees or above to measure human capital. However, even the people with college degrees or above who really play a role in innovation are the people who are employed in the innovation or R&D departments. Therefore, the measurement of talent in this paper mainly refers to scientific research practitioners. Referring to the practice of Lai and Ji (2015) [29], the proportion of the regional employment structure composed of scientific research talent is first selected, and the regional employment structure of talent is represented by the proportion of sci-

entific research practitioners in each region among that year's national scientific research practitioners. Secondly, the adjustment factor of talent allocation is determined according to the industry "relative monopoly degree" proposed by Jin (2005) [58]; the ratio of regional scientific research practitioners to local practitioners is taken as the talent allocation fitting indicator. Referring to the practice of the China Economic Growth Frontier Research Group (2014) [59], the basic adjustment factor (x) of talent allocation is formed by dividing the fitting index by the proportion of the region's output added value in the national GDP. Finally, the talent allocation variable is expressed by the product of the proportion of the regional talent employment structure and the adjustment factor. The specific calculation method is as follows:

$$talalloc_{i,t} = \frac{talent_{i,t}}{\sum\limits_{i=1}^{258} talent_{i,t}} \times \left( \frac{talent_{i,t}}{L_{i,t}} \div \frac{gdp_{i,t}}{\sum\limits_{i=1}^{258} gdp_{i,t}} \right) \tag{2}$$

where $talalloc_{i,t}$ is the talent allocation level in region $i$ during period $t$, $talent_{i,t}$ is the number of talented people in region $i$ during period $t$, $L_{i,t}$ is the number of employees in region $i$ during period $t$, and $gdp_{i,t}$ is the gross product of region $i$ in period $t$.

### 3.2.3. Mechanism Variables

Innovation drive (*innodr*): It is difficult for the development mode of factor-input-driven economic growth to ensure high-quality economic development. To achieve high-quality economic development, the economic development mode needs to be shifted from factor-driven to innovation-driven development. The lack of innovation in various regions will hinder high-quality development, so high-quality development is constrained by the innovation level. Talented people, as high-end knowledge owners, are also the key element in achieving innovation goals, and the accumulation of innovation-driven talent will inevitably lead to the reconfiguration of talent among regions. The measurement of innovation drive cannot be directly estimated, and the urban innovation index (The data are from the 2017 report on China's Urban and Industrial Innovation Capacity issued by the Industrial Development Research Center of Fudan University) can indirectly reflect the innovation level of the region; therefore, this paper used the urban innovation index in the 2017 report on China's Urban and Industrial Innovation Power issued by the Industrial Development Research Center of Fudan University to measure the innovation drive level of each region, and missing values were supplemented by interpolation.

### 3.2.4. Mechanism Variables

Industrial agglomeration level (*indclu*): On the one hand, industrial agglomeration can produce a scale effect and externality, reduce enterprise production costs and transaction costs, and improve economic growth efficiency. On the other hand, it can form a "labor pool" to provide talent support for technological innovation and form an endogenous growth momentum, thus promoting high-quality economic development. The location entropy index can accurately reflect the spatial distribution of industries geographically; therefore, this paper used the location entropy index of secondary industry practitioners to measure the level of industrial agglomeration in each region.

Marketization (*market*): Promoting the effective flow of talent in different industries, departments, and regions is key to improving enterprise efficiency. The improvement of the marketization level creates favorable conditions for the cross-regional flow of talent, promotes the flow of talent to departments with higher production efficiency, and further optimizes the regional allocation of talent. This paper refers to the practice of Fan et al. (2019) [60] and measures the level of regional marketization from five aspects: the relationship between the government and the market; the development of the non-state-owned economy; the growth degree of product markets; the growth degree of factor markets; and the market service environment. This study also uses the principal component analysis method to determine the weight of each basic indicator and then synthesizes the

marketization indicators. The details of the five indicators of marketization are shown in Table 2 below.

**Table 2.** Marketization index system.

| Indicator Dimension | Measurement Method |
| --- | --- |
| Relationship between government and market | Proportion of regional fiscal expenditure in local GDP |
| Development of non-state-owned economy | Proportion of the total number of private and individual employees in the number of local employees |
| Product market development | The number of local enterprises |
| Growth degree of factor market | Proportion of foreign direct investment in GDP |
| Market service environment | Proportion of business service employees in the number of employees |

Remuneration incentive (*incent*): Remuneration is an important factor for talent to consider when looking for jobs. Therefore, the salary level is particularly obvious in a number of policies introduced by various regions to attract talent, which have also set off a surge of talent flow. The allocation of talent between regions is therefore affected. However, the salary level is a burden for local people, and whether it can improve the allocation of talent between regions in the long run remains to be tested. This paper used the regional per capita wage level to measure the salary incentive in various regions, and then examined the role of salary incentives in the allocation of talent between regions.

Financial autonomy (*fiauto*): Government decisions often lead the direction of economic development in various regions. Under the existing performance appraisal system, local governments may disregard the quality of economic development in pursuit of temporary achievements, and the choice of government decisions is often greatly affected by finance. Therefore, it is necessary to include the government's financial autonomy in the analysis framework. This paper referred to the practice of Yu et al. (2019) [61], using the ratio of revenue within the financial budget to expenditure within the financial budget to measure the financial autonomy of local governments.

Financial development level (*finance*): Financial development can give full play to the allocation function of financial resources, guide financial institutions and social capital to increase financial support for green projects, curb capital investment in highly polluting and energy-consuming industries, and ensure that green energy conservation and environmental protection enterprises obtain sufficient funds, thus promoting green economic development. This paper refers to the practice of Xie (2019) [19], using the year-end CNY deposit balance of financial institutions to measure the financial development level of each city.

*3.3. Data Description*

This paper took 258 cities in China as the research objects, and the period of 2004–2019 as the observation cycle. The data were obtained from the China Statistical Yearbook, China Population and Employment Statistical Yearbook, China Urban Statistical Yearbook, China Provincial and Municipal Economic Development Yearbook, and the statistical yearbooks of various provinces and cities. Among them, a few missing data were supplemented by interpolation. In order to exclude the impact of economic price changes on the statistical data, some indicators were converted using GDP deflators with 2000 as the base period in the calculation process. In addition, the multiple collinearity test using the variance expansion factor found that all VIF values were less than 10. Therefore, there was no serious multicollinearity problem between the variables. Descriptive statistics of the variables are shown in Table 3.

**Table 3.** Descriptive statistics of variables.

| Variable | Obs | Mean | Std.Dev. | Min | Max | Cor(Y,X) |
|---|---|---|---|---|---|---|
| *ecogrq* | 4128 | 0.364 | 0.140 | 0.004 | 0.936 | 1 |
| *talalloc* | 4128 | 0.020 | 0.040 | 0.000 | 0.443 | 0.333 *** |
| *innodr* | 4128 | 14.478 | 74.478 | 0.005 | 2073.283 | 0.314 *** |
| *indclu* | 4128 | 0.899 | 0.361 | 0.002 | 1.844 | 0.151 *** |
| *market* | 4128 | 0.069 | 0.050 | 0.004 | 0.451 | 0.433 *** |
| *incent* | 4128 | 40,700.72 | 22,398.29 | 6207.11 | 321,000 | 0.296 *** |
| *fiauto* | 4128 | 0.493 | 0.224 | 0.026 | 1.541 | 0.598 *** |
| *finance* | 4128 | $5.58 \times 10^7$ | $1.41 \times 10^8$ | 1,040,000 | $2.38 \times 10^9$ | 0.458 *** |

Note: *** $p < 0.01$.

## 4. Empirical Results and Analysis

### 4.1. Benchmark Regression

Since the value range of high-quality development, the explained variable in this paper, is between 0 and 1, the data are truncated. If ordinary least squares (OLS) is selected to regress the regional talent allocation and high-quality economic development, it will cause a serious error in the parameter estimation. Tobit models belong to truncated regression models, which can effectively remedy the defects of the OLS regression method. Therefore, this paper used a restricted dependent variable regression model—the tobit model—to empirically examine the impact of regional talent allocation on high-quality economic development, and ensured the stability of the impact of the core explanatory variables through gradual return [62]. The regression results are shown in Table 4.

**Table 4.** Benchmark regression results.

| Variable | (1) ecogrq | (2) ecogrq | (3) ecogrq | (4) ecogrq | (5) ecogrq | (6) ecogrq |
|---|---|---|---|---|---|---|
| *talalloc* | 0.039 *** (0.002) | 0.043 *** (0.002) | 0.036 *** (0.002) | 0.035 *** (0.001) | 0.034 *** (0.001) | 0.032 *** (0.001) |
| *var(e.ecogrq)* | 0.017 *** (0.000) | 0.017 *** (0.000) | 0.014 *** (0.000) | 0.014 *** (0.000) | 0.010 *** (0.000) | 0.010 *** (0.000) |
| *indclu* | | 0.078 *** (0.009) | 0.056 *** (0.008) | 0.042 *** (0.008) | −0.070 *** (0.007) | −0.059 *** (0.007) |
| *market* | | | 1.218 *** (0.042) | 1.081 *** (0.044) | 0.216 *** (0.043) | 0.100 ** (0.045) |
| *incent* | | | | 0.031 *** (0.003) | 0.046 *** (0.003) | 0.022 *** (0.004) |
| *fiauto* | | | | | 0.155 *** (0.004) | 0.141 *** (0.004) |
| *finance* | | | | | | 0.018 *** (0.002) |
| Constant | 0.551 *** (0.008) | 0.573 *** (0.008) | 0.459 *** (0.008) | 0.137 *** (0.036) | 0.161 *** (0.030) | 0.092 *** (0.031) |
| Observations | 4128 | 4128 | 4128 | 4128 | 4128 | 4128 |

Note: figures in brackets are robust standard errors; ** $p < 0.05$, and *** $p < 0.01$.

Column (1) of Table 4 shows the regression results when only considering the influence of the regional allocation of talent on high-quality economic development. The regression results show that the regression coefficient of the regional allocation of talent for high-quality economic development is 0.039, and it passes the 1% significance level test. After adding other variables one by one in columns (2) to (6), the symbol and significance of the regression coefficient of regional allocation of talent for high-quality economic development do not change significantly. This means that the regional allocation of talent can effectively promote the high-quality development of the regional economy, and this conclusion is stable. This is mainly because, as high-end knowledge owners, the improvement of the regional talent allocation level is not only conducive to heterogeneous knowledge

exchange, promoting the exchange and the sharing of knowledge innovation, creation, and technical experience, but is also able to effectively alleviate the efficiency gap caused by the distortion of regional talent allocation, thus injecting new momentum into environmental protection and high-quality economic development. The higher the talent allocation level, the more conducive it is to the improvement of output efficiency; both the level of economic development and the quality of economic development can play a significant role.

Column (2) shows the regression result of adding the industrial agglomeration variable. It can be seen from the table that the coefficient of industrial agglomeration for high-quality economic development is 0.1078, and it passes the 1% significance level test, indicating that industrial agglomeration can effectively promote high-quality economic development. This is mainly because industrial agglomeration can enhance exchanges and cooperation between enterprises, improve the regional division of labor, and bring scale effects and externalities. At the same time, industrial agglomeration is also conducive to the centralized treatment of pollutants, reducing the environmental loss caused by economic development and promoting high-quality economic development.

From the perspective of the control variables, the regression coefficient of marketization for high-quality economic development is significantly positive in the regression results of column (3) after the addition of marketization variables, indicating that the improvement of the marketization level can effectively promote high-quality economic development. This is mainly because the promotion of marketization is conducive to promoting the orderly flow of various factors, breaking the resource solidification, and improving the resource mismatch. At the same time, the improvement of the level of marketization can also curb the vicious competition among economic entities and stimulate market vitality. This not only effectively curbs the occurrence of efficiency loss caused by rent seeking, but also improves the efficiency of the market-oriented allocation of resources, thereby improving the output efficiency of economic entities. Therefore, the improvement of the level of marketization can effectively promote high-quality economic development.

Column (4) shows the regression result after adding the salary incentive variable. The result shows that the regression coefficient of salary incentive for high-quality economic development is 0.031, and it passes the 1% significance level test, indicating that improving salary incentives can effectively promote high-quality economic development. This is mainly because strengthening the salary incentive intensity can mobilize the enthusiasm of employees to improve the output efficiency and reduce the waste of resources.

Column (5) shows the regression result after considering the government's financial autonomy. The result shows that the regression coefficient of the government's financial autonomy for high-quality economic development is 0.155, and it passes the 1% significance level test, indicating that the improvement of financial autonomy can effectively promote high-quality economic development, which is mainly because the improvement of financial autonomy can provide financial support for the government's decision making and is conducive to the promotion and implementation of technology development and environmental governance decisions. In the stage of high-quality development, all local governments have turned their development goals towards innovative development. The improvement of financial autonomy can provide financial support for the introduction of talent from outside, their internal training, and the guidance of local industrial transformation, which is conducive to regional innovation and development. Therefore, the improvement of financial autonomy can effectively promote regional environmental protection and high-quality economic development.

Column (6) shows the regression result after considering the financial development level. The result shows that the regression coefficient of the financial development level for high-quality economic development is 0.018, and it passes the 1% significance level test, indicating that the improvement of the financial development level can effectively promote the improvement of the high-quality economic development level. This is mainly because the improvement of the financial development level can give full play to the allocation function of financial resources by guiding financial institutions and social capital to increase

financial support for green projects and restraining capital investment in industries with high pollution and high energy consumption, meaning that green energy conservation and environmental protection enterprises can obtain sufficient funds and continue to expand reproduction.

### 4.2. Endogenous Treatment and Robustness Test

From the above analysis, we can see that regional talent allocation can effectively promote high-quality economic development. In the process of transforming the economic growth model from factor-input-driven to innovation-driven growth, in order to achieve the goal of high-quality economic development with environmental protection as the core, various regions are also actively introducing talent, and high-quality development will also force the improvement of regional talent allocation. Based on this, it was speculated that there may be some endogenous problems in the econometric model. Therefore, the two-stage least squares (2SLS) method was used to re-estimate equation (1) by seeking the instrumental variables, and the stepwise regression method was used to ensure the stability of the impact of the core explanatory variables. With regard to the selection of tool variables, considering that the regional talent allocation level lagging behind by one period is closely related to the regional talent allocation level of the current period, and that it is difficult to have an impact on the high-quality growth of the current period, the selection of the regional talent allocation lagging behind by one period as a tool variable can not only effectively solve the time lag problem, but also effectively solve the endogenous problem. Table 5 shows the estimation results.

**Table 5.** Regression results of endogenous treatment.

| Variable | (1) | (2) | (3) | (4) | (5) | (6) |
|---|---|---|---|---|---|---|
| | *ecogrq* | *ecogrq* | *ecogrq* | *ecogrq* | *ecogrq* | *ecogrq* |
| *talalloc* | 0.041 *** | 0.041 *** | 0.036 *** | 0.036 *** | 0.042 *** | 0.038 *** |
| | (0.002) | (0.002) | (0.002) | (0.002) | (0.002) | (0.002) |
| *indclu* | | 0.007 *** | 0.008 *** | 0.015 *** | −0.002 | −0.002 |
| | | (0.002) | (0.002) | (0.002) | (0.002) | (0.002) |
| *market* | | | 1.247 *** | 1.060 *** | 0.263 *** | 0.121 *** |
| | | | (0.048) | (0.048) | (0.047) | (0.046) |
| *incent* | | | | 0.047 *** | 0.038 *** | 0.012 ** |
| | | | | (0.004) | (0.003) | (0.005) |
| *fiauto* | | | | | 0.145 *** | 0.132 *** |
| | | | | | (0.005) | (0.005) |
| *finance* | | | | | | 0.020 *** |
| | | | | | | (0.003) |
| Constant | 0.565 *** | 0.570 *** | 0.462 *** | −0.020 | 0.272 *** | 0.181 *** |
| | (0.011) | (0.011) | (0.010) | (0.038) | (0.035) | (0.036) |
| Observations | 3870 | 3870 | 3870 | 3870 | 3870 | 3870 |
| R-squared | 0.132 | 0.135 | 0.290 | 0.316 | 0.501 | 0.512 |

Note: figures in brackets are robust standard errors; ** $p < 0.05$, and *** $p < 0.01$.

Through the analysis of the regression results in Table 5, it was found that, after using tool variables to re-estimate the impact of regional talent allocation on high-quality economic development, the regression results of regional talent allocation on high-quality economic development had not changed both in the coefficient sign and significance level compared with the benchmark regression results in Table 4, which indicates that, after using tool variables to re-estimate the regional talent allocation and high-quality economic development, the conclusion that regional talent allocation can effectively promote high-quality economic development is still a robust one.

### 4.3. Mechanism Inspection

From the previous analysis, it can be seen that the regional allocation of talent plays a role in promoting high-quality economic development. Theoretical analysis shows

that the regional allocation of talent promotes high-quality economic development by promoting innovation. To verify this assumption, the following intermediary effect model was constructed, and then the above mechanism analysis was empirically tested:

$$innodr_{i,t} = \beta_0 + \beta_1 talalloc_{i,t} + \beta_2 controls + \lambda_i + \phi_t + \varepsilon_{i,t} \tag{3}$$

$$ecogrq_{i,t} = \eta_0 + \eta_1 innodr_{i,t} + \theta_3 controls + \lambda_i + \phi_t + \varepsilon_{i,t} \tag{4}$$

$$ecogrq_{i,t} = \theta_0 + \theta_1 talalloc_{i,t} + \theta_2 innodr_{i,t} + \theta_3 X_{i,t} + \theta_4 controls + \lambda_i + \phi_t + \varepsilon_{i,t} \tag{5}$$

In the above formula, $ecogrq_{i,t}$ refers to the level of high-quality economic development, $talalloc_{i,t}$ is the level of regional talent allocation, $innodr_{i,t}$ is the level of regional innovation driven by mechanism variables, and $X_{i,t}$ is the interaction between regional talent allocation and innovation drive; other variables are the same as above. The mechanism inspection results are shown in Table 6.

**Table 6.** Mechanism inspection results.

| Variable | (1) innodr | (2) ecogrq | (3) ecogrq | (4) ecogrq | (5) innodr | (6) ecogrq | (7) ecogrq | (8) ecogrq |
|---|---|---|---|---|---|---|---|---|
| talalloc | 0.130 *** (0.014) | | 0.034 *** (0.001) | 0.028 *** (0.001) | 0.090 *** (0.009) | | 0.038 *** (0.002) | 0.031 *** (0.002) |
| innodr | | 0.013 *** (0.002) | 0.004 * (0.002) | 0.038 *** (0.004) | | 0.015 *** (0.002) | 0.005 ** (0.002) | 0.039*** (0.005) |
| Talalloc × innodr | | | | 0.007 *** (0.001) | | | | 0.007 *** (0.001) |
| indclu | −0.077 *** (0.007) | −0.001 (0.002) | −0.001 (0.001) | −0.001 (0.001) | −0.052 *** (0.012) | −0.002 (0.002) | −0.002 (0.002) | −0.001 (0.002) |
| market | 0.491 (0.331) | 0.161 *** (0.049) | 0.127 *** (0.045) | 0.076 * (0.044) | 0.627 ** (0.319) | 0.155 *** (0.049) | 0.118 ** (0.046) | 0.072 (0.048) |
| incent | 0.705 *** (0.047) | −0.019 *** (0.004) | 0.009 ** (0.004) | 0.011 *** (0.004) | 0.561 *** (0.035) | −0.022 *** (0.005) | 0.009 * (0.005) | 0.010 ** (0.005) |
| fiauto | 0.083 ** (0.041) | 0.104 *** (0.004) | 0.127 *** (0.004) | 0.122 *** (0.004) | 0.334 *** (0.031) | 0.103 *** (0.005) | 0.130 *** (0.005) | 0.125 *** (0.005) |
| finance | 1.028 *** (0.034) | 0.025 *** (0.004) | 0.017 *** (0.004) | 0.012 *** (0.003) | 1.183 *** (0.018) | 0.022 *** (0.004) | 0.015 *** (0.004) | 0.010 *** (0.004) |
| var(e.ecogrq) | | 0.011 *** (0.000) | 0.010 *** (0.000) | 0.009 *** (0.000) | | | | |
| Constant | −23.703 *** (0.189) | 0.210 *** (0.068) | 0.237 *** (0.063) | 0.274 *** (0.062) | −24.812 *** (0.247) | 0.286 *** (0.072) | 0.304 *** (0.069) | 0.335 *** (0.068) |
| Observations | 4128 | 4128 | 4128 | 4128 | 3870 | 3870 | 3870 | 3870 |
| R-squared | 0.877 | | | | 0.878 | 0.424 | 0.512 | 0.528 |

Note: figures in brackets are robust standard errors; * $p < 0.1$, ** $p < 0.05$, and *** $p < 0.01$.

Column (1) in Table 6 shows the regression result of regional talent allocation for innovation drive. The result shows that the regression coefficient of regional talent allocation for innovation drive is 0.130, and it passes the 1% significance level test, indicating that regional talent allocation can effectively promote the regional innovation level. Column (2) shows the regression result of innovation drive for high-quality economic development. The result shows that the regression coefficient of innovation drive for high-quality economic development is 0.013, and it passes the 1% significance level test, indicating that innovation drive can effectively promote high-quality economic development. At the same time, the estimated coefficients of regional talent allocation and innovation drive in the estimation results of column (3) pass the 10% significance level test at the least, and the coefficients are positive, indicating that innovation is a mechanism variable of regional talent allocation that affects high-quality economic development.

The fourth column shows the regression result after adding the interactive item of regional talent allocation and innovation drive. After adding the interactive item of regional talent allocation and innovation drive, the regression coefficient of regional talent allocation and innovation drive for high-quality economic development is still positive, and both pass the 1% significance level test. In particular, the size and significance of the regression coefficient of innovation drive for high-quality economic development are further improved after adding the interaction term of the two, indicating that regional talent allocation can effectively enhance the impact of innovation drive on high-quality economic development. This is mainly because regional talent allocation and innovation drive can interact. Regional talent allocation can effectively promote the improvement of the regional innovation level, and the improvement of the innovation level, in turn, can effectively promote the improvement of regional talent allocation. Therefore, the combined effect of the two can not only promote regional talent allocation and innovation-driven benign development, but also play a significant role in promoting high-quality economic development.

Columns (5) to (8) in Table 6 show the regression results of the endogenous treatment and robustness test of the intermediary model using the two-stage least squares method with the lag periods of regional talent allocation and innovation drive as the tool variables. Comparing the robustness test results with the benchmark regression results, it was found that the impact of regional talent allocation on innovation-driven development, the impact of innovation-driven development on high-quality economic development, and the impact of both of them on high-quality economic development are consistent with the benchmark regression results, indicating that the conclusions drawn from the previous mechanism regression are robust; that is, regional talent allocation can promote high-quality economic development by promoting regional innovation.

*4.4. Heterogeneity Analysis*

The above analysis reflects the overall effect of regional talent allocation and innovation drive on high-quality economic development. However, China is a vast country, and there is a big difference between its economic development level, urban hierarchy, regional characteristics, resource endowment, and economic growth potential. The regional talent allocation level of economically underdeveloped regions is also far from that of developed cities; therefore, it is necessary to further investigate whether there is heterogeneity in the impact of regional talent allocation and innovation drive on economic quality. Based on this, this paper divided the research sample into three subsamples of east, central, and west according to the regions of the 258 cities, and into urban agglomeration samples and non-urban agglomeration samples according to the list of urban agglomerations listed in the 14th Five-Year Plan Outline, so as to investigate the heterogeneity of regional talent allocation and innovation drive in high-quality economic development. The regression results are shown in Tables 7 and 8.

In Table 7, the first column shows the regression result of regional talent allocation and innovation drive in the eastern region for high-quality economic development; the second column shows the regression result of regional talent allocation and innovation drive in the central region for high-quality economic development; and the third column shows the regression result of regional talent allocation and innovation drive in the western region for high-quality economic development. Columns (4) to (6) show the regression results of the endogenous treatment and robustness test using the two-stage least squares method with the lag periods of regional talent allocation and innovation drive as the tool variables. From the regression results, we can see whether the regional allocation of talent, innovation drive, and the interaction of the regional allocation of talent and innovation drive play a significant role in promoting high-quality economic development in the three regions.

**Table 7.** Regression results by region.

| Variable | (1) ecogrq | (2) ecogrq | (3) ecogrq | (4) ecogrq | (5) ecogrq | (6) ecogrq |
|---|---|---|---|---|---|---|
| talalloc | 0.018 *** | 0.031 *** | 0.043 *** | 0.018 *** | 0.036 *** | 0.047 *** |
| | (0.002) | (0.003) | (0.002) | (0.003) | (0.003) | (0.003) |
| innodr | 0.037 *** | 0.071 *** | 0.052 *** | 0.036 *** | 0.081 *** | 0.055 *** |
| | (0.006) | (0.008) | (0.006) | (0.008) | (0.010) | (0.009) |
| talalloc × innodr | 0.005 *** | 0.015 *** | 0.009 *** | 0.006 *** | 0.017 *** | 0.009 *** |
| | (0.001) | (0.001) | (0.001) | (0.001) | (0.002) | (0.002) |
| indclu | 0.001 | −0.007 *** | 0.003 | 0.000 | −0.008 *** | 0.002 |
| | (0.002) | (0.002) | (0.003) | (0.003) | (0.002) | (0.003) |
| market | 0.127 ** | −0.192 * | 0.183 ** | 0.133 * | −0.202 * | 0.164 * |
| | (0.064) | (0.109) | (0.089) | (0.069) | (0.117) | (0.092) |
| incent | 0.004 | 0.015 ** | 0.049 *** | 0.008 | 0.017 * | 0.048 *** |
| | (0.008) | (0.007) | (0.008) | (0.011) | (0.009) | (0.009) |
| fiauto | 0.156 *** | 0.101 *** | 0.129 *** | 0.158 *** | 0.102 *** | 0.134 *** |
| | (0.009) | (0.007) | (0.006) | (0.014) | (0.008) | (0.009) |
| finance | 0.011 | 0.016 *** | −0.018 *** | 0.012 | 0.011 | −0.023 *** |
| | (0.007) | (0.006) | (0.006) | (0.008) | (0.007) | (0.007) |
| var(e.ecogrq) | 0.008 *** | 0.010 *** | 0.008 *** | | | |
| | (0.000) | (0.000) | (0.000) | | | |
| Constant | 0.312 ** | 0.171 * | 0.440 *** | 0.260 * | 0.267 ** | 0.561 *** |
| | (0.121) | (0.101) | (0.099) | (0.151) | (0.109) | (0.105) |
| Observations | 1536 | 1568 | 1024 | 1440 | 1470 | 960 |
| R-squared | | | | 0.553 | 0.448 | 0.640 |

Note: figures in brackets are robust standard errors; * $p < 0.1$, ** $p < 0.05$, and *** $p < 0.01$.

**Table 8.** Regression results of urban agglomerations and non-urban agglomerations.

| Variable | (1) ecogrq | (2) ecogrq | (3) ecogrq | (4) ecogrq |
|---|---|---|---|---|
| talalloc | 0.039 *** | 0.023 *** | 0.041 *** | 0.025 *** |
| | (0.002) | (0.002) | (0.002) | (0.002) |
| innodr | 0.083 *** | 0.031 *** | 0.091 *** | 0.031 *** |
| | (0.006) | (0.004) | (0.008) | (0.005) |
| talalloc × innodr | 0.017 *** | 0.004 *** | 0.018 *** | 0.004 *** |
| | (0.001) | (0.001) | (0.001) | (0.001) |
| indclu | −0.001 | −0.001 | −0.002 | −0.001 |
| | (0.002) | (0.002) | (0.002) | (0.002) |
| market | 0.084 | −0.003 | 0.092 | −0.010 |
| | (0.083) | (0.053) | (0.089) | (0.056) |
| incent | 0.032 *** | −0.014 ** | 0.031 *** | −0.016 ** |
| | (0.006) | (0.006) | (0.008) | (0.006) |
| fiauto | 0.093 *** | 0.139 *** | 0.097 *** | 0.139 *** |
| | (0.006) | (0.006) | (0.007) | (0.006) |
| finance | −0.003 | 0.018 *** | −0.009 | 0.018 *** |
| | (0.005) | (0.004) | (0.006) | (0.004) |
| var(e.ecogrq) | 0.010 *** | 0.008 *** | | |
| | (0.000) | (0.000) | | |
| Constant | 0.312 *** | 0.411 *** | 0.446 *** | 0.434 *** |
| | (0.092) | (0.082) | (0.108) | (0.081) |
| Observations | 1744 | 2384 | 1635 | 2235 |
| R-squared | | | 0.481 | 0.602 |

Note: figures in brackets are robust standard errors; ** $p < 0.05$, and *** $p < 0.01$.

Through comparison, it can be found that the regression coefficient of regional talent allocation for high-quality economic development has increased in the three regions, and this conclusion is still valid in the regression results of the endogenous treatment and robustness test in columns (4) to (6), indicating that the marginal effect of regional talent

allocation on high-quality economic development shows the characteristics of an increasing marginal effect from east to west. This may be because, among the three regions, the eastern region has the highest talent allocation level, followed by the central region, while the western region has the lowest talent allocation level. As the core variable of high-quality economic development, the role of regional talent allocation in high-quality economic development may be restricted by the law of diminishing marginal effect. Therefore, the marginal effect of regional talent allocation on high-quality economic development increases from east to west. This shows that improving the level of regional talent allocation can more effectively promote the high-quality economic development of the western region. The effect of innovation drive and the interaction of regional talent allocation and innovation drive on high-quality economic development is the strongest in the central region, followed by the western region, while it is the weakest in the eastern region. This conclusion is still valid in the regression results of the endogenous treatment and robustness test in columns (4) to (6), which may be because the role of innovation drive in high-quality economic development is also restricted by the law of diminishing marginal effect. The spillover effect of innovation driven high-quality economic development in the eastern and western regions can affect the high-quality economic development of the central region. Therefore, the marginal effect of high-quality economic development driven by innovation is generally the strongest in the central region, followed by the western region, and is the weakest in the eastern region. The marginal effect of innovation on high-quality economic development can be summarized as follows: the central region is the strongest, the western region is the second strongest, and the eastern region is the weakest.

Columns (1) and (2) in Table 8 show the regression results of regional talent allocation and innovation drive in the non-urban agglomeration samples and urban agglomeration samples for high-quality economic development. Columns (3) and (4) show the regression results of the endogenous treatment and robustness test using the two-stage least squares method with the lag periods of regional talent allocation and innovation drive as the tool variables. From the table, it can be seen that, regardless of whether it concerns regional talent allocation, innovation drive, or the interaction of regional talent allocation and innovation drive, the impact on high-quality economic development shows a significant role in promoting non-urban agglomerations and urban agglomerations. Through the comparison of the non-urban and urban agglomerations, it can be found that, regardless of whether it concerns the regional allocation of talent, innovation drive, or the interaction of regional talent allocation and innovation drive, non-urban agglomerations have a higher marginal effect on high-quality economic development. This conclusion is still valid in the regression results of the endogenous treatment and robustness test in columns (3) and (4). This shows that the marginal contribution of improving regional talent allocation and the innovation drive level to the high-quality development of non-urban agglomerations' economy is stronger.

*4.5. Analysis of Nonlinear Effects Based on Innovation Drive*

Although the regional allocation of talent has a positive role in promoting high-quality economic development, the intensified competition between regions caused by innovation drive will cause talent flow. When the spillover effect of talent flow is greater than the negative impact, it is conducive to promoting the regional allocation of talent to play a role in high-quality economic development. When the innovation drive causes talent who do not need to flow to participate in the flow under external impact, it will place talent in the adaptation period of the new environment, resulting in a low efficiency of regional talent allocation; this will also make it difficult for regional talent allocation to play an effective role in high-quality economic development. Based on this, it is speculated that there is an innovation drive threshold effect of regional talent allocation on the development of a

high-quality economy. Therefore, Equation (5) was transformed into a nonlinear threshold model, and the specific design is as follows (6):

$$\begin{aligned}
ecogrq_{i,t} = \beta_0 &+ \beta_1 talalloc_{i,t} + \beta_2 talalloc_{i,t} * I(innodr_{i,t} \le \eta) \\
&+ \beta_3 talalloc_{i,t} * I(innodr_{i,t} > \eta) + \beta_4 controls + \varepsilon_{i,t}
\end{aligned} \tag{6}$$

where $innodr_{i,t}$ is the threshold variable, $\eta$ is the threshold value to be estimated, $I(\bullet)$ is the indicator function, and the other variables are consistent with Equation (2). According to the relative size of the threshold variable $innodr_{i,t}$ and the threshold value $\eta$, the sample was divided into several subsamples, and the difference between regions is reflected in the difference in the parameter sum of $\beta_2$ and $\beta_3$.

Therefore, the threshold regression method was further used to test whether there is a threshold effect of innovation drive through regional talent allocation on high-quality economic development. Table 9 shows the inspection results.

**Table 9.** Threshold effect test.

| Threshold | RSS | MSE | F-stat | Prob | Crit10 | Crit5 | Crit1 |
|---|---|---|---|---|---|---|---|
| Single | 7.5444 | 0.0018 | 96.21 | 0.0020 | 27.8806 | 35.4015 | 56.3596 |
| Double | 7.4678 | 0.0018 | 42.16 | 0.0260 | 24.3168 | 30.9856 | 66.9679 |
| Triple | 7.4314 | 0.0018 | 20.17 | 0.2160 | 25.0818 | 30.3068 | 41.5676 |

It can be seen in Table 9 that innovation drive fails to pass the triple threshold test, while both the single threshold and double threshold tests are passed at the 5% significance level at least. This shows that there is a dual threshold effect driven by innovation in the impact of regional talent allocation on high-quality economic development. The likelihood ratio function sequence LR($\gamma$) can better fit the trend of the threshold parameters and more intuitively reflect the threshold value of the threshold variables and their corresponding confidence intervals. It can be seen in Figure 1 that when the likelihood ratio LR($\gamma$) is 0, the double threshold values of innovation drive are 0.4898 and 10.2214; the area below the dotted line is the confidence interval of the threshold value at the 95% level.

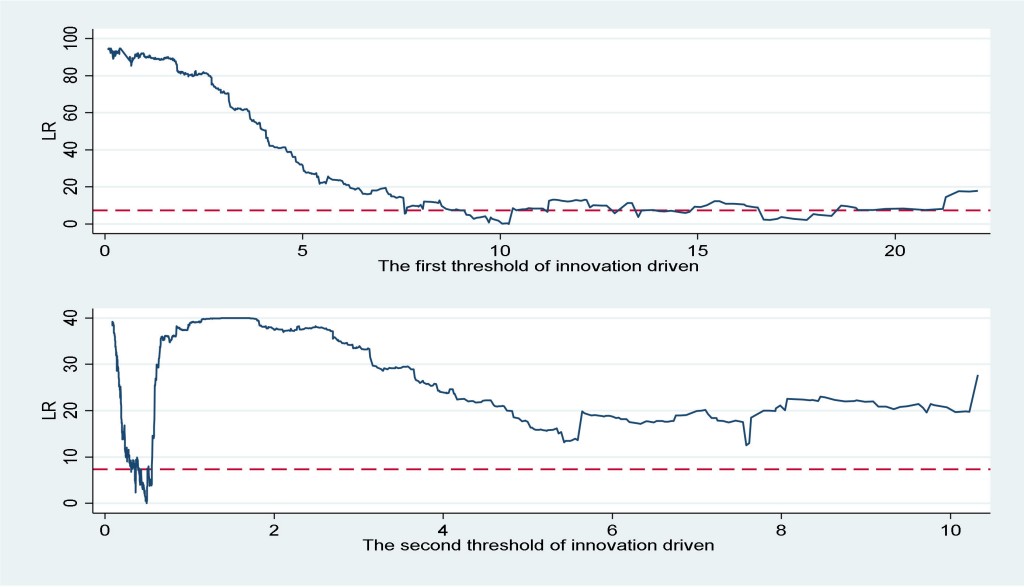

**Figure 1.** Threshold values and confidence intervals of innovation drive. Note: The curve is the value of the threshold variable, and the red line is a statistical value of 7.35.

After the threshold value driven by innovation was obtained using the bootstrap method, the whole sample was divided into different intervals with the threshold value as the critical point, and then the impact of regional talent allocation on high-quality economic

development with the change in the threshold value was estimated. The estimation results are shown in Table 10.

**Table 10.** Double threshold regression results.

| Variable | (1) | (2) |
| --- | --- | --- |
| | Coefficient | t |
| *talalloc* | 0.288 *** | |
| *Innodr* < 0.4898 | (0.099) | 2.92 |
| *talalloc* | −0.306 *** | |
| 0.4898 < *innodr* < 10.2214 | (0.054) | −5.62 |
| *talalloc* | 0.107 ** | |
| 10.2214 < *innodr* | (0.059) | 1.81 |
| *indclu* | 0.0004 | |
| | (0.001) | 0.62 |
| *market* | 0.027 *** | |
| | (0.004) | 6.73 |
| *incent* | 0.141 *** | |
| | (0.032) | 4.45 |
| *fiauto* | −0.002 | |
| | (0.005) | −0.42 |
| *finance* | 0.019 *** | |
| | (0.003) | 5.99 |
| Constant | 0.072 *** | |
| | (0.018) | 3.89 |
| Obs | 4128 | |
| F | 71.76 | |

Note: figures in brackets are robust standard errors; ** $p < 0.05$, and *** $p < 0.01$.

It can be seen in Table 10 that when the innovation drive level is lower than 0.4898, the estimated coefficient of regional talent allocation for high-quality economic development is 0.288, and it passes the 1% significance level test. When the innovation drive level is between 0.4898 and 10.2214, the estimated coefficient of regional talent allocation for high-quality economic development is −0.306, and it passes the 1% significance level test. When the innovation drive level is higher than 10.2214, the estimated coefficient of regional talent allocation for high-quality economic development is 0.107, and it passes the 5% significance level test. This shows that when the innovation drive level is between 0.4898 and 10.2214, it is not conducive to the effect of regional talent allocation on high-quality economic development. This is mainly because innovation drive can cause talent flow and thus affect regional talent allocation. When the innovation drive level is between 0.4898 and 10.2214, it may cause talent who do not need to flow to participate in the flow under external impact, leading to talent being placed in the adaptation period of the new environment; the spillover effect of talent flow will be less than the negative impact of talent flow, which will lead to a low efficiency of regional talent allocation, and it will be difficult for regional talent allocation to effectively play its role in high-quality economic development. To sum up, there is a nonlinear threshold effect of regional talent allocation on high-quality economic development due to the impact of the innovation drive intensity.

## 5. Conclusions

Under the background of relying on innovation to drive high-quality economic development, optimizing the effective spatial allocation of talent is an urgent problem to be solved to achieve high-quality economic development with the goal of environmental protection. This paper took innovation drive as the starting point to reveal the impact mechanism of regional talent allocation on high-quality economic development, and used the panel data of 258 cities in China from 2004 to 2019 to empirically test the impact of regional talent allocation on high-quality economic development. This study found that the improvement of the regional talent allocation level can effectively promote high-quality

economic development. The mechanism test shows that the regional allocation of talent can promote the high-quality development of the economy by influencing regional innovation, and the joint effect of the regional allocation of talent and innovation-driven growth can also significantly promote the high-quality development of the economy. The heterogeneity test found that the impact of regional talent allocation on high-quality economic development showed a law of an increasing marginal effect from east to west, while the effect of innovation drive and the interaction between regional talent allocation and innovation drive on high-quality economic development was the strongest in the central region, followed by the western region, and was weakest in the east. This study also found that both the regional allocation of talent and the innovation-driven impact on high-quality economic development in non-urban agglomeration cities have a higher marginal effect than those in urban agglomeration cities, and there is a double threshold effect of innovation-driven regional talent allocation on high-quality economic development. When the innovation drive is between 0.4898 and 10.2214, the spillover effect of innovation-driven talent flow is less than the negative impact of talent flow, which is not conducive to the role of the effect of regional talent allocation in high-quality economic development.

Based on the conclusion that regional talent allocation can promote high-quality economic development by influencing regional innovation, we believe that it is necessary to strengthen innovation-driven leadership, build a talent promotion platform, and enhance regional talent competitiveness. Guided by innovation drive, targeted talent training and investment should be carried out, and the interests of the educated should be explored based on the direction of industrial development, so as to achieve "teaching students according to their aptitude", thus promoting the accumulation of human capital and promoting the sustainable development of the economy. Secondly, based on the impact of salary incentives and financial autonomy on high-quality economic development, we believe that we should improve the quality of talent training in universities and strengthen investment in education. Education is the basis of talent generation, and the length of education directly determines the talent quantity and quality. Without education, there is no talent supply. In view of the high externality of education, relying solely on personal investment can easily lead to an insufficient supply of education and even an unfair distribution of it. Therefore, the government needs to intervene in the supply of education, including the investment and regulation of education. Thirdly, based on the impact of marketization on high-quality economic development, it is proposed that the talent flow mechanism be improved and that a talent governance system guided by market demand be built. To ensure the smooth flow of talent market channels, by improving the talent flow mechanism, building a talent management system, managing the chaos of "talent introduction", strengthening the guidance of talent support policies, and attracting more high-level talent, so as to create a "talent chain, industry chain, and entrepreneurship chain", and to realize the organic integration of the three chains, we can ensure the reasonable and orderly flow of talent and realize the reasonable and effective allocation of talent resources. Finally, based on the impact of industrial agglomeration and financial development level on high-quality economic development, we propose to accelerate industrial restructuring, build resource agglomeration platforms supported by finance, attract talent into advantageous industries, rely on talent to support industrial development, and form a virtuous interaction between industrial structure optimization and talent mobility. By improving the matching degree between talent and industrial structure, we can avoid the waste of human resources caused by 'underemployment of the highly skilled and over-employment of the less skilled', thus enhancing the efficiency of human capital allocation.

**Author Contributions:** Conceptualization, L.L. and S.S.; methodology, L.L.; software, L.L. and J.L.; validation, L.L. and S.S.; formal analysis, L.L.; investigation, L.L. and S.S.; resources, S.S.; data curation, L.L.; writing—original draft preparation, L.L.; writing—review and editing, L.L. and S.S.; visualization, L.L.; supervision, S.S.; project administration, S.S.; funding acquisition, S.S. and J.L. All authors have read and agreed to the published version of the manuscript.

**Funding:** This study was supported by the Key Projects of Anhui Provincial Department of Education (2022AH050575) and the 2022 Anhui Social Science Innovation and Development Research Project "Research on Regional Allocation and Effects of Talent in Anhui Province under the Background of High Quality Development".

**Institutional Review Board Statement:** Not applicable.

**Informed Consent Statement:** Not applicable.

**Data Availability Statement:** The data presented in this study are available upon request from the corresponding author. The data are not publicly available due to the research data subjects.

**Conflicts of Interest:** The authors declare that they have no conflict of interest.

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
