# Peer review of "Research on the Effect of Regional Talent Allocation on High-Quality Economic Development—Based on the Perspective of Innovation-Driven Growth"

_sustainability, doi:10.3390/su15076315_

Round 1
Reviewer 1 Report
Remarks:
1. In general, the article does not meet the formatting requirements of the Sustainability journal. Therefore, it is recommended that the article be proofread by a native speaker.
2. The abstract does not include the purpose of the study.
3. The use of panel data of 258 cities in China from 2004 to 2019 is outdated. It is now the end of 2022.
4. The use of 2015 and 2019 and other older references in the introduction seriously reduces the rationale for the relevance and importance of the study.
5. I don't see scientific hypothesis testing in the article.
6. Sections 5.1 and 5.2 lack evidence-based and cost-effective recommendations. Without the above sentence, the sections are declarative and not credible.
Author Response
Thank you for your guidance on this paper. We have revised the paper according to your comments, which not only makes our paper more clear and accurate in language, but also makes the content of our manuscript fuller. Thank you again.

Reviewer 2 Report
This article meets the conditions to be published in the journal, being a topic of interest to the scientific community for sustainable development studies. A content review should be conducted to summarize the presentation of the hypothesis defense.
Author Response
Thank you for your approval of the article we submitted, and we revised it based on the final review results of the editorial department, which made our paper not only clearer and more accurate in language, but also fuller in content. Thank you again.

Reviewer 3 Report
From a methodological point of view, this is an interesting contribution that brings a different dimension to the Innovation driven analysis.
Author Response
Thank you for your approval of the article. We further revised it according to the editorial department's review results, making it more normative. Thank you again.

Reviewer 4 Report
Information provided on talent allocation, economic growth, and environmental quality in introduction section is mix. Describe title sequentially in gap(s) analysis form in introduction section.
Literature for the development of H1 on page 4 is not sufficient and that relevant. Explain under the rubric of well-known theory(ies) that how regional talent allocation promote regional innovation and thus improve environmental quality and economic development.
What support H2? (non-linear relationship) explain!
Measures mentioned in table 4 need be explained in literature.
Author Response
Thank you for your guiding comments on this paper. We have revised the paper with reference to your comments, making our paper not only clearer and more accurate in language, but also fuller in content. Thank you again.

Reviewer 5 Report
Your manuscript could make an interesting contribution, but it will require quite a lot of work. In its current state, the manuscript is hard to read and understand, and you should look for guidance concerning the writing. Your reasoning needs to be much clearer, as do the key messages. Sentences might be tightened to eliminate ambiguity.
Examples of significant errors:
Change "Innovation Driven" to "Innovation-Driven" or the entire title to "Regional allocation of talents and high-quality economic development: perspectives on innovation."
Change "goals such as the environment" to "environmental sustainability goals."
"…the supply of talents in the secondary industry is increasing under-supply" -- I can’t tell what the authors intended.
Firstly, the focus towards the relationship between the allocation of talents, high-quality development, and green transformation was not clear whether this was a hypothesis applied to the panel data or whether this was a hypothesis of prior studies. Was this a finding or a specific lens through which the panel data was examined? The impact on green transformation is neither immediately clear nor easily understood.
Secondly, the research question is not strong enough to attract readers’ curiosity. It is almost certain that increase in and allocation of talents will help boost high-quality developments. If the answer to the research question is obvious, then there is little need for argument. The reasoning in research did not help to flesh out the scientific idea.
Thirdly, specific technical phrases are used throughout, often with little explanation. This made the manuscript hard to follow and did not help to justify the research process. The introduction should be curated to stress the novelty of the study - the meaning of terms introduced to the context of the research must be explained once again, but with elaboration. Such terms include 'high-quality development, 'green development', and 'gradual liberalization of the labor market'.
Fourthly, I would suggest a re-organization of the material. In particular, the paragraphs became convoluted at circumstances with Discussions presenting in the Results and Conclusions. This is partly the reason for my first remark (since the distinction is not clear, the gap between what the literature already achieves and what is hypothesized is not fully justified). I expect the improved structure and clarity would strengthen the piece as "the discoveries as presented by the literature" and the authors' conclusions making a stronger statement of contribution. Then you compare the findings of your research with those of the following cornerstone articles:
The Allocation of Talent: Implications for Growth, The Quarterly Journal of Economics
The Allocation of Talent and U.S. Economic Growth, Econometrica
Author Response

(The authors gave the same response as above.)

Round 2
Reviewer 4 Report
Satisfactory. Minor English changes required.
Author Response
Thank you for pointing out that there are some problems in the expression of the article due to the differences in mother tongue. We have adopted your opinion and selected English native language experts to proofread the article.

Reviewer 5 Report
Major comments:
The text still reads like a post-graduate student's homework after the major revision. I must admit that the historical explanation of economic development taught me a lot, but how does this matter relate to sustainability?
The first few sentences of Section 1 are verbatim repetitions of the opening few sentences of the abstract. Please refrain from doing so throughout the entire manuscript. In fact, more elaborations on methodology should take the place of a significant portion of the introductions.
How do you address the alleged issue of data endogeneity in statistical registers?
The discussions and conclusions are still quite general. I would suggest an in-depth exploration of the findings, going into the details about the model, and describing the significance of your contributions in light of relevant sources.
I would suggest illustrating statistical hypotheses in a conceptual framework. When illustrating statistical hypotheses in a conceptual framework, this means that the researcher is visually representing their ideas about how the hypothesis relates to other variables in the study. This can be done using diagrams or flowcharts.
Minor comments:
Line 13, page 1: …has raised discussions revolving around environmental protection requirements.
Line 15, page 1: … local governments have prioritized the development model to a green, sustainable format
Line 21, page 1: use spatial allocation of talents instead of allocation of talents in space
Line 22, page 1: …and to facilitate the benign interaction… (since you have used realize right before)
Line 31 -32, page: please rephrase. You are repeating yourself.
Please include the contribution of the paper to the literature at the end of the abstract.
